# Pathogenic Germline Mutations of DNA Repair Pathway Components in Early-Onset Sporadic Colorectal Polyp and Cancer Patients

**DOI:** 10.3390/cancers12123560

**Published:** 2020-11-28

**Authors:** Pi-Yueh Chang, Shih-Cheng Chang, Mei-Chia Wang, Jinn-Shiun Chen, Wen-Sy Tsai, Jeng-Fu You, Chia-Chun Chen, Hsiu-Ling Liu, Jy-Ming Chiang

**Affiliations:** 1Department of Laboratory Medicine, Chang Gung Memorial Hospital, No. 5 Fu-Shing St. Kweishan, Taoyuan 33305, Taiwan; changpy@adm.cgmh.org.tw (P.-Y.C.); changsc137@cgmh.org.tw (S.-C.C.); ottermika@cgmh.org.tw (M.-C.W.); s48633001@cgmh.org.tw (H.-L.L.); 2Department of Medical Biotechnology and Laboratory Science, Chang Gung University, No. 259, Wenhua 1st Rd., Kweishan, Taoyuan 33302, Taiwan; 3Department of Colorectal Surgery, Chang Gung Memorial Hospital, No. 5 Fu-Shing St. Kweishan, Taoyuan 33305, Taiwan; chenjs@cgmh.org.tw (J.-S.C.); wensyt@gmail.com (W.-S.T.); you3368@cgmh.org.tw (J.-F.Y.); 4Molecular Medicine Research Center, Chang Gung University, No. 259, Wenhua 1st Rd., Kweishan, Taoyuan 33302, Taiwan; chenchiachun@gap.cgu.edu.tw

**Keywords:** early onset, colorectal cancer, cancer susceptibility gene, germline mutation, polyp, normal control

## Abstract

**Simple Summary:**

Colorectal cancer (CRC) screening by immuno-fecal occult blood tests (iFOBTs) begins at age 50 in average-risk persons. However, the incidence of early-onset CRC has risen; of the cases, at least half are sporadic CRC without a family history. The authors of this study found a high percentage of de novo germline mutation in young sporadic CRC patients, as well as in sporadic colorectal polyp and control groups. All the mutated genes contribute to various DNA-repair pathways, hinting that a loss of genomic integrity play a crucial role in the development of CRC. The early identification of cancer-susceptible individuals by multigene panels in younger individuals who may be missed under current iFOBT screening could contribute to preventing CRC.

**Abstract:**

Given recent increases in the proportion of early-onset colorectal cancer (CRC), researchers are urgently working to establish a multi-gene screening test for both inherited and sporadic cancer-susceptible individuals. However, the incidence and spectrum of germline mutations in young sporadic CRC patients in East Asian countries and, especially, in sporadic polyp carriers and normal individuals are unknown. Peripheral blood samples were collected from 43 colonoscopy-proved normal controls and from 50 polyp patients and 49 CRC patients with no self-reported family history of cancer. All participants were under 50 years old. Next-generation sequencing with a panel of 30 CRC-associated susceptibility genes was employed to detect pathogenic germline mutations. The germline mutation carrier rates were 2.3%, 4.0%, and 12.2% in the normal, polyp, and cancer groups, respectively. A total of seven different mutations in six DNA repair pathway-related genes (*MLH1, BRCA1, BRCA2, CHEK2, BLM,* and *NTHL1*) were detected in nine participants. One frameshift mutation in *BRCA2* and one frameshift mutation in the *CHEK2* gene were found in a normal control and two colorectal polyp patients, respectively. One young sporadic CRC patient carried two heterozygous mutations, one in *MLH1* and one in *BRCA1*. Three mutations (MLH1 p.Arg265Cys, MLH1 p.Tyr343Ter and CHEK2 p.Ile158TyrfsTer10) were each found in two independent patients and were considered “founder” mutations. This is the first report to demonstrate high percentage of germline mutations in young sporadic colorectal polyp, CRC, and general populations. A multi-gene screening test is warranted for the proactive identification of cancer-predisposed individuals.

## 1. Introduction

International guidelines recommended the fecal occult blood test (FOBT) and colonoscopy as screening tools for colorectal cancer (CRC), with a starting age of 50 years old. Most countries in Europe, the Americas, and East Asia have implemented the screening program with annual or biennial FOBTs, followed by colonoscopy when FOBT results are positive. However, due to the variation in financial resources and participation rates in different countries, the cancer detection rate varies greatly from 0.29 (Thailand) to 5.9 (Netherland) per 1000 participants [1]. In Taiwan, the National Health Administration provides free immuno-fecal occult blood tests (iFOBTs) every two years for people aged 50–74 for screening of CRC since 2004 [2]. This nationwide screening program has been successful, reportedly reducing CRC mortality in five million Taiwanese people by 62% when compared to screened and unscreened populations [3].

However, although the incidence of colorectal cancer has been on a steady decline in recent decades, the proportion of early-onset CRC defined by diagnosis before 50 years old appears to be on an increasing trend in the US and Europe [4,5]. Rebecca L. Siegel investigated the CRC incidence patterns in the United States from 1974 to 2013. They found the CRC incidence was increasing among young adults with a net annual increase of 4% compared to a net annual decrease of 2% for those aged 75 years and older [5]. In Taiwan, according to the cancer registration statistics in 2017 [6], approximately 12.5% of CRC patients were younger than 50 years of age and, of them, 48.9% were diagnosed in stages III and IV versus 41.6% in the CRC group >50 years old. The current screening strategy does not cover this younger subgroup.

Early-onset colorectal cancer patients are commonly defined as those with a diagnosis at younger than 50 years of age. Based on the presence of family history, two subtypes have been found: the “sporadic” and “inherited” subtypes. The former is diagnosed without any family history, while the latter diagnosis includes a well-defined family history. Clinically, young individuals with hereditary CRC syndromes may be alert to this situation and more likely to undertake frequent early screening. In contrast, young patients without any family cancer history have little opportunity to be detected until they are diagnosed with advanced CRC [7]. The awareness of clinicians and the appropriateness of their screening strategy should be frequently reviewed and updated.

In general, it is known that germline mismatch repair (MMR) gene mutations, together with *APC* gene mutations, contribute significantly to inherited CRC [8]. However, whether a similar positive rate of germline mutations can be observed in sporadic early-onset CRC has been unclear. Additionally, multi-gene detection could serve as a surrogate tool for cancer screening in the young population. Indeed, the development of next-generation sequencing (NGS) techniques enabled researchers to reveal that a substantial proportion of young sporadic CRC patients harbor germline mutations [9,10,11,12,13]. However, these studies have varied in the genes screened by NGS, and the prevalence and spectrum of pathogenic germline mutations in designated young sporadic CRC patients is still unknown.

Here, we designed a 30-gene panel including conventional CRC-predisposing genes and pleiotropic non-CRC-associated cancer susceptibility genes (e.g., *BRCA2* and *TP53*) to explore the germline mutation rate and type in young CRC patients without a family history, and then we compared that information among CRC patients [14,15], polyp patients, and colonoscopy-proven negative individuals. Our goal was to clarify the molecular entities and pathways underlying the development of CRC and to provide proof-of-concept for a cost-effective genetic tool intended for the large-scale screening of the young population. Our results could also impact the further development of family-tailored prevention and clinical management strategies.

## 2. Results

### 2.1. 2.3%, 4.0%, and 12.2% of Germline Mutation Rate Found in Normal, Polyp, and CRC Groups

A total of 142 young patients were enrolled in this study, including 43 colonoscopy-proven negative normal control individuals, 50 colorectal polyp patients, and 49 CRC patients. Pathogenic germline mutations were found in one normal control participant, two patients in the polyp group, and six patients in the sporadic CRC group. The clinical features and germline mutation carrier rates are summarized in Table 1. Age and gender exhibited even distributions across the three groups. However, we observed gradual elevations of the serum carcinoembryonic antigen (CEA) value and germline mutation carrier rate (2.3%, 4.0%, and 12.2%, respectively) sequentially through the normal, polyp, and cancer groups.

### 2.2. Mutations in Early-Onset CRC and Polyp Patients Occurred Mainly in DNA Repair Pathway-Related Genes

Only pathogenic and likely pathogenic variants annotated by the National Center for Biotechnology Information (NCBI) ClinVar or VarSome were analyzed. A total of seven different mutations distributed across six genes (*MLH1*, *BRCA1*, *BRCA2*, *CHEK2*, *BLM*, and *NTHL1*) were detected in nine patients. One young sporadic CRC patient carried heterozygous mutations in both *MLH1* and *BRCA1*. Three mutations (MLH1 p.Arg265Cys, MLH1 p.Tyr343Ter, and CHEK2 p.Ile158TyrfsTer10) were each found in two independent patients, and these were designated “founder” mutations. The detailed distribution data and information on the germline mutations in each group are presented in Table 2 and Appendix A, respectively. Interestingly, all six of the altered genes encode factors involved in the DNA repair pathway, including components that contribute to mismatch repair (*MLH1)*, recombinant repair (*CHEK2, BRCA1, BRCA2,* and *BLM*), and base-excision repair (*NTHL1*). *MLH1* was the most frequently mutated gene, as it was found to be mutated in four early-onset sporadic CRC patients (44.4% or 4/9), and *CHEK2* (22% or 2/9) was found to be mutated in two polyp patients.

### 2.3. Phenotypic Characteristics of Mutation Carriers in the Early-Onset Sporadic CRC Group

As shown in Table 3, we compared the clinico-pathologic features between the mutation carriers and non-mutation carriers of the 49 young sporadic CRC patients. We found that germline mutation carriers presented a signet ring cell histology significantly more than adenocarcinoma or mucinous (100% or 2/2 vs. 8.5% or 4/47). Though the differences were not statistically significant, mutation carriers also tended to have poorer-differentiated, later-stage, and larger-sized tumors in the right colon.

## 3. Discussion

Many early-onset CRC patients have inherited cancer syndromes or a family history of cancer in at least one first-degree relative, and they thus might harbor mutations in cancer-susceptibility genes. The frequency of germline mutations in early-onset CRC patients has been reported to be around 10–20% [9,10,11,12,13]. However, these studies recruited different proportions of patients with and without a family history of CRC. To date, few investigations have explored the germline mutation rate in early-onset CRC patients without a family history [16]. In the present study, we found pathogenic germline mutations in CRC-associated genes among 12.2% of early-onset sporadic CRC patients, 4% of sporadic polyp patients, and 2.3% of normal controls.

Table 4 summarizes the results of four previous studies reporting the frequency of germline mutations in early-onset CRC patients or all-age CRC. If we focus on patients without a family history, the germline mutation rate in these studies was only around 6–10%. However, this may have been caused by a deduction of the number of individuals who were assumed to be familial with so-called “inherited” mutations on MMR-associated genes and familial adenomatous polyposis genes. In the present study, if we subtract the patients who carried only *MLH1* mutations (*n* = 3), the mutation detection rate of 6.1% (3/49) in sporadic early-onset CRC was similar to those found in the previous investigations. However, does this truly represent the mutation distribution among young CRC patients without a family history? The de novo mutation of the human genome is an important event that contributes to the generation of the genetic diversity needed for evolution [17]. It has also been shown to be a major cause of early-onset genetic disorders and malignancies. For example, 2.3% of Lynch syndrome patients without a family history were found to have de novo MMR gene mutations [18]. Twenty-six familial adenomatous polyposis (FAP) patients with 15 putative de novo *APC* mutations that may arise during the meiosis have also been reported [19]. An *MLH1* c.666dupA de novo mutation identified in a 31-year-old colorectal cancer patient and an *APC* c.694C>T de novo mutation identified in a Chinese family were random evidence for de novo mutation events in an assumed “inherited” subtype of CRC [20,21]. Therefore, despite our inability to enroll the probands’ family members to prove that the presumed de novo mutations were only detected in probands but not in their parents or grandparents, the independent cancer event on probands that was observed in the three-generation pedigree trees of six early-onset CRC patients indicated the possibility of de novo mutation (Appendix A). The 12.2% de novo germline mutation rate in MMR genes and other non-CRC cancer susceptibility genes (*BRCA1, BLM,* and *NTHL1*) reported herein for the young sporadic CRC group is reasonable and expectable.

Sporadic colorectal polyps are often considered a precursor of cancer, and the removal of suspicious polyps under colonoscopy is recommended to reduce their risk of further developing to cancer. However, recurrent polyps have been observed in 13.8% of patients at first year post-polypectomy and in 60% of patients at third year post-polypectomy [22]. This indicates that germline mutation background may contribute to this phenotype, in addition to the influence of food and lifestyle. However, only germline mutations detected in inherited polyposis patients with burdens of >10 polyps have been reported [23]. We are the first group to investigate the prevalence and spectrum of germline mutation in sporadic polyp patients with less than five polyps. The same *CHEK2* mutation of c. 472 delA was present in two independent polyp patients in our study and was thus designated as a suspect “founder” mutation in Taiwanese group. This novel *CHEK2* variant has not previously been reported in polyp patients, but the *CHEK2* mutation had been found to be common in other sporadic CRC groups (Table 4). This indicates the possible development of polyps that carried the *CHEK2* mutation to future malignance transformation if not removed at that time.

This study was also the first to explore the germline mutation frequency in a normal population. One 38-year-old male participant with a normal colonoscopy and no self-reported family history of cancer harbored a pathogenic *BRCA2* frameshift mutation. According to a review published by Sopik [24], women younger than 50 years with a *BRCA1* mutation had a five-fold increased risk of CRC when compared to that in women without mutation. The CRC risk for men carrying the *BRCA2* mutation is unknown, but such men likely warrant increased cancer surveillance.

Interestingly all of germline mutations found in early-onset sporadic CRC and polyp patients were related to the DNA damage repair pathway [25]. In previous studies (Table 4), the top six gene alterations found in non-inherited CRC were monoallelic changes in *MUTYH, ATM, CHEK2, BRCA2*, *BRCA1*, and *BLM* (specifically in a Chinese population). All these genes act as gatekeepers and respond to DNA damage for the maintenance of genome integrity. The identified variants were dominantly or biallelically inherited with varied penetrance. In theory, defects in these repair genes will result in the slow and progressive somatic accumulation of pathogenic variants (two-hit theory) [26]. This will increase the risk of cancer in multiple tissues that are vulnerable to this type of damage, such as those with a rapid turnover rate (e.g., colonic and extra-enteric cells). In this study, we did not observe other malignancies instead of colorectal cancers among our patients. However, if the follow-up period is long enough, extracolonic malignancies should be monitored carefully.

Carriers with germline alterations in *BRCA1/2*, *BLM*, *CHEK2*, and *NTHL1* have all been reported to be at an elevated risk for developing early-onset CRC with unique histological characteristics [27,28,29,30]. In our study, young sporadic CRC patients harboring mutations in these genes exhibited more frequent signet ring cell adenocarcinoma and tended to have more advanced and larger tumors in the right side colon (Table 3). Right side colon predominance was one of the clinical characteristics of lynch syndrome with mismatch repair gene mutations, while familial colorectal cancer type X (FCCTX) with unknown mutations have left side or distal colon predominance [31]. This feature is concordant with our findings in which four of six sporadic early-onset CRC patients carrying *MLH1* germline mutations.

Ideally, a population-based genetic cancer-screening protocol should be comprehensive and effectively cover most affected candidate genes. Conventional phenotype- and syndrome-directed CRC risk assessment is only suitable for familial CRC. Some investigators have recommended that germline multi-gene testing should only be applied for family members with positive microsatellite instability (MSI)/MMR immunohistochemistry results in their tumor [32] in order to relieve the family’s uncertainty and anxiety. However, in this study, we demonstrate that the incidence of germline mutations in young sporadic CRC patients is comparable to that seen in inherited CRC. Recently, targeted therapy against members of the DNA repair pathway, such as the inhibition of poly ADP ribose polymerase (PARP), has been reported to benefit patients with mutations in homologous repair genes [33]. Therefore, the proactive identification of pathogenic germline mutations in DNA repair-relevant genes could improve the population-level prevention of CRC, as well as its genetic counseling and surveillance.

## 4. Materials and Methods

### 4.1. Study Subjects and Sample Collection

The three cohorts comprised a total of 142 Taiwanese participants, and all of them were admitted to Chang Gung Memorial Hospital for medical care or health examination services during 2016–2019. All participants were aged 50 years or less and lacked any reported family cancer history in their first-, second-, or third-degree relatives. The accuracy and completeness of the patient-reported family history data were verified when possible through the careful review of medical records by a senior doctor. The first cohort included 49 sporadic CRC patients. The demographic features of the cancer patients, such as tumor stage, size, location (tumors at the cecum, ascending colon, hepatic flexure, and transverse colon were classified as right-side tumors; tumors at the splenic flexure, descending colon, and sigmoid colon were classified as left-side tumors; and those at the rectum were classified separately), histology, grade, and recurrence status were all recorded. The second cohort contained 50 individuals whose health checks revealed 1–5 polyps that were removed during colonoscopy. Upon histological examination, 94% of the polyps were found to be hyperplastic or adenomatous, while the remaining 6% were tubulovillous. The third cohort contained 43 normal controls collected from a clinical health center, all of whom had negative colonoscopy findings. All patients and healthy individuals provided written informed consent, and the study was approved by the institutional review board of Chang Gung Memorial Hospital (case number: 103-7047B, 201801201B0, 201900596B0A3). Whole blood was placed into 3 mL EDTA tubes from all participants before their surgical or colonoscopy procedure. Genomic DNA was extracted using a QIAamp DNA Blood Mini Kit (Qiagen, Hilden, Germany). Each DNA sample was checked for purity using a NanoDrop (Thermo Fisher Scientific, Waltham, MA, USA) and for concentration using a Qubit 2.0 Fluorometer (Life Technologies, Carlsbad, CA, USA).

### 4.2. Design of the 30 CRC-Susceptibility Gene Panel

To assemble our NGS CRC-susceptibility gene panel, we referred to the following sources: the National Comprehensive Cancer Network (NCCN) Guidelines Insights for Genetic/Familial High-Risk Assessment: Colorectal, Version 3.2017 [34]; the guidelines of the UK Cancer Genetics Group [35], which provides a consensus for genes to be included in cancer panel tests offered by UK genetics services; and large-scale academic or commercial review articles [8,36]. The panel included 30 genes. Of them, 13 genes (*MLH1, MSH2, MSH6, PMS2, EPCAM, TP53, MLH3, CHEK2, CDH1, ATM, BRCA1, BRCA2*, and *RPS20)* were related to non-polyposis syndrome, 10 genes (*STK11, PTEN, BMPR1A, SMAD4, GREM1, RNF43, BLM, GALNT12, AKT1*, and *PIK3CA*) were related to non-adenomatous polyposis diseases, and seven genes *(APC, MUTYH, POLE, POLD1, NTHL1, AXIN2*, and *CTNNA1)* were related to adenomatous polyposis syndrome. The Ion AmpliSeq™ Designer v4.2.4 cloud-based software program (Life Technologies) was used to design the customized panel. This panel consisted of 699 amplicons (target length: 125–375 bp) in two pools; it covered the entire exonic regions and 25-bp padding sequences at the exon/intron junctions for a total screened sequence of 190 kb.

### 4.3. Next-Generation Sequencing and Analysis Pipeline

AmpliSeq multiplexed libraries were constructed using an Ion AmpliSeq Library Kit 2.0 according to the manufacturer’s protocol (Applied Biosystems, Life Technologies, Carlsbad, CA, USA). The AMPure bead (Beckman Coulter, Brea, CA, USA)-purified libraries were assessed for their concentrations and size distributions using a Qubit 2.0 Fluorometer (Life Technologies, Carlsbad, CA, USA) and an Agilent Bioanalyzer 2100 with a high-sensitivity DNA chip (Agilent Technologies, Santa Clara, CA, USA). Quantified and barcoded libraries were diluted to 180 pM and pooled together. Template preparation and chip loading were performed using Ion Chef (Life Technologies). Enriched Ion Sphere Particles were sequenced using an Ion S5 Plus system on an Ion 520^TM^ Chip that consisted of 10 samples.

Raw sequencing data were trimmed of barcoded adapter sequences and filtered for poor signal reads, and then they were aligned to the human genome build 19 reference genome (hg19). Variant calling was done through the platform-specific pipeline of “VariantCaller v5.10” (Life Technologies). Advanced variant annotation was facilitated by uploading the Variant Call Format (VCF) file from Variant Caller to the cloud software package Ion Reporter (Thermo Fisher Scientific) and the web-based software wANNOVAR (Wang Genomics Lab, http://wannovar.wglab.org/). Variants were manually inspected for authenticity using the Integrative Genomics Viewer (IGV) and filtered-in by the following criteria: variant located in an exonic or splicing site, variant with nonsynonymous or frameshift or stop–gain effect, and an allele frequency of less than 1% in 1000 Genomes (1000G), the Exome Aggregation Consortium (ExAC), and the Exome Sequencing Project (ESP6500) population databases. Finally, only variants interpreted as pathogenic and likely pathogenic by the NCBI ClinVar (https://www.ncbi.nlm.nih.gov/clinvar/) or VarSome (https://varsome.com/) databases, which follow the guidelines of the American College of Medical Genetics and Genomics (ACMG) [37], were considered for further evaluation. All filter-in variants were confirmed by Sanger sequencing. The assay design and analysis workflow are depicted in Figure 1.

### 4.4. Statistical Analysis

Descriptive statistics are summarized as percentages, means, and standard deviations. Between-group comparisons were conducted using Student’s t-test, one-way ANOVA, and Pearson’s chi-squared for each marker. A *p*-value of less than 0.05 (two-tailed) was considered statistically significant. All statistical analyses were conducted using SPSS Statistics 22 (SPSS Inc., Armonk, NY, USA).

## 5. Conclusions

We herein report the first study to explore the prevalence and spectrum of early-onset sporadic CRC and polyp patients in a Taiwanese population. This study had several limitations, such as the rather low number of enrolled affected patients and the fact that the risk factors and tumor behavior of colorectal cancer in an Asia population may have some differences with Western countries [38]; therefore, the findings in this study should be carefully applied. However, the high incidence of ethnicity-specific germline mutation in young Taiwanese CRC and polyp patients suggests that it may be possible to design a cost-effective genetic test to probe cancer-predisposing candidates and identify younger at-risk individuals who may be missed under the current iFOBT screening policy.

## Figures and Tables

**Figure 1 cancers-12-03560-f001:**
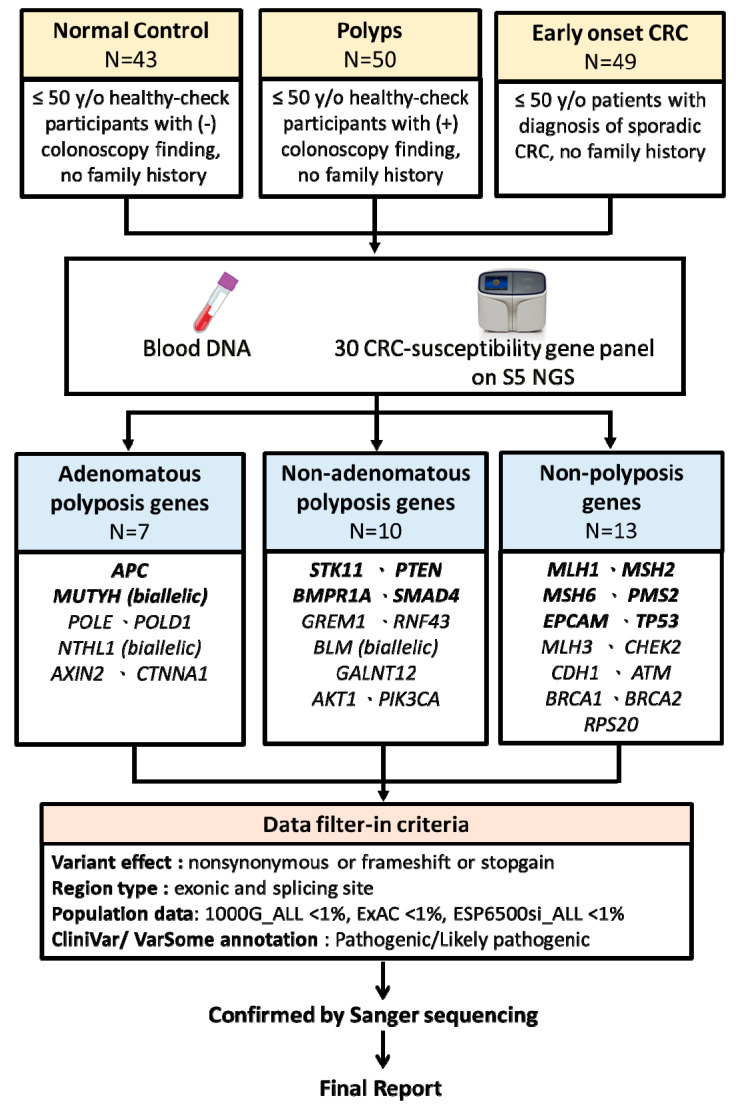
Assay design and the next-generation sequencing analysis pipeline.

**Table 1 cancers-12-03560-t001:** Demography and germline mutation carrier rate in three cohorts. CRC: colorectal cancer. CEA: carcinoembryonic antigen.

Features	Normal (*n* = 43)	Polyps (*n* = 50)	CRC (*n* = 49)	*p*-Value
Gender (F/M)	25/17	25/25	24/25	0.635
Age (Mean ± SD)	41.4 ± 6.8	42.2 ± 5.9	43.4 ± 5.6	0.287
CEA (Mean ± SD)	1.04 ± 0.78	1.23 ± 1.10	11.47 ± 19.46	<0.005
Mutation carrier no (%)	1(2.3%)	2 (4%)	6 (12.2%)	0.105

**Table 2 cancers-12-03560-t002:** Distribution of mutation gene and type among normal, polyp, and CRC groups.

Involved Pathway	Mutation Gene	Mutation Type	Normal (*n* = 43)	Polyps (*n* = 50)	CRC (*n* = 49)
Mismatch Repair	*MLH1*	p.Arg265Cys; p.Tyr343Ter (X 2)	0	0	3
Recombinational Repair	*CHEK2*	p.Ile158TyrfsTer10 (X 2)	0	2	0
	*BRCA2*	p.Asp885ArgfsTer3	1	0	0
	*BLM*	p.Phe1189LeufsTer10	0	0	1
Base-excision Repair	*NTHL1*	p.Ser116ArgfsTer38	0	0	1
Combined	*MLH1* and *BRCA1*	p.Arg265Cys and p.Gln1577Ter	0	0	1
Total mutation			1 (2.3%)	2 (4%)	6 (12.2%)

Note 1: Underline (X 2), double underline, and wave line (X 2) highlight the same mutation on two individuals. Note 2: The nomenclature of mutation is followed by Human Genome Variation Society (HGVS) international standard.

**Table 3 cancers-12-03560-t003:** Clinico-pathologic features of 49 young sporadic CRC patients between mutation carriers and non-mutation carriers. TNM: tumor size; lymphatic involvement and presence of metastases.

Features	All	Mutation Carriers	Non-Mutation Carrier	*p* Value
Patient No	49	6	43	
Age	43.4 ± 5.7	43.7 ± 5.3	43.4 ± 5.8	0.906
Gender				
Female	24	3 (13%)	21 (87%)	0.957
Male	25	3 (12%)	22 (88%)	
Tumor TNM stage				
1	8	0 (0%)	8 (100%)	0.237
2	10	1 (10%)	9 (90%)	
3	21	2 (9.5%)	19 (90.5%)	
4	10	3 (30%)	7 (70%)	
Tumor Location				
Right colon	10	3 (30%)	7 (70%)	0.055
Left colon and rectum	39	3 (7.7%)	36 (92.3%)	
Tumor size				
<5 cm	32	2 (6.3%)	30 (93.7%)	0.079
≥5 cm	17	4 (24%)	13 (76%)	
Tumor histology				
Adenocarcinoma	44	4 (9%)	40 (91%)	0.001
Mucinous	3	0 (0%)	3 (100%)	
Signet ring cell	2	2 (100%)	0 (0%)	
Tumor grade				
Well	7	0 (0%)	7 (100%)	0.334
Moderate	34	4 (12%)	30 (88%)	
Poor	8	2 (25%)	6 (75%)	
Pre_operation CEA				
<5 ng/mL	31	2 (6.5%)	29 (93.5%)	0.104
≥5 ng/mL	18	4 (22%)	14 (78%)	
Recurrence (except TNM 4)				
Yes	8	0 (0%)	8 (100%)	0.360
No	31	3 (10%)	28 (90%)	

**Table 4 cancers-12-03560-t004:** Summary of finding in studies using next generation sequencing (NGS) and target gene panel for detection of germline mutation in early onset CRC or unselected CRC groups. LS: lynch syndrome; FAP: familial adenomatous polyposis

Author [Reference] (Published Year/Nation)	Number of Patients	Inclusion Criteria	% Inherited CRC	Detection Platform (Gene Contents)	Positive Rate of Pathogenic Mutations in Total Patients	Positive Rate of Pathogenic Mutations in Non-Inherited CRC Group	Mutation Gene in Non-Inherited CRC (Number of Patients)
Pearlman, R. et al. [12] (2017/USA)	450	<50 y/o	11% (37 LS and 11 FAP)	NGS (25-gene panel)	16% (72 of 450)	6% (24 of 402)	Monoallelic *MUTYH* (7), *APC* c.3920T>A (3), *ATM* (3), *BRCA2* (4), *BRCA1* (2), *PALB2* (2), *CDKN2A* (1), *SMAD4* (1), *ATM/CHEK2* (1),
Yurgelun, M.B. et al. [13] (2017/USA)	1058	Unselected CRC (at all age, only 31.8% diagnosed before 50 y/o)	3.9% (33 LS and 8 FAP)	NGS (25-gene panel)	9.9% (105 of 1058)	6.4% (65 of 1017)	Monoallelic *MUTYH* (18), *APC* c.3920T>A (14), *BRCA2* (8), *ATM* (10), *BR1P1* (3), *BRCA1* (3), *PALB2* (2), *NBN* (2) *CHEK2* (2), *CDKN1A* (1), *TP53* (1), *BRAD1* (1)
Stoffel, E.M. et al. [9] (2018/USA)	430	<50 y/o	57.2% (any relatives with CRC)	NGS (154-gene panel)	18.4% (79 of 430)	7.2% (13 of 181)	Monoallelic *MUTYH* (8), *SMAD4* (2), *BRCA1* (1), *TP53* (1), *CHECK2* (1)
Gong, R. et al. [10] (2019/China)	618	Unselected CRC (at all age, only 48.7% diagnosed before 50 y/o)	44.7% (any relatives with CRC)	NGS (73-gene panel)	18.1% (112 of 618)	10.2% (35 of 342)	*ATM* (5), *CHEK2*(4), *BLM* (4), *TSHR* (4), *FANCA/CC* (4), *BARD1* (3), *BR1P1* (2), *BRCA1*/*2* (2), monoallelic *MUTYH* (1), *TP53* (1) and other 5 genes

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
