# Peer review of "Pathogenic Germline Mutations of DNA Repair Pathway Components in Early-Onset Sporadic Colorectal Polyp and Cancer Patients"

_cancers, 2020, doi:10.3390/cancers12123560_

Round 1

Reviewer 1 Report

Dear authors,

    I would like to congratulate you for the paper, for the idea to integrate new methods of early diagnostic in one of the most challenging and frequent oncologic pathology.

Colorectal cancer is 3rd most frequent cancer in both male and female population, with a decreasing incidence in the last decade determined by introduction of the screening protocol with colonoscopy in all population over 50 years. Still, colorectal cancer in younger population started being more frequent, imposing an identification strategy for this tumor behavior.

This observational study compared 3 cohorts of Asian Population with normal colon mucosa, colon polyp and colorectal cancer from the mutational point of view through Next Generation Sequencing panel trying to identify a common mutation that could determine the malignant transformation.

Minor concerns;

  1. Please specify what is the international screening protocol and mention why in Taiwan the protocol is different, cite and mention comparative studies in terms of impact on the incidence.
  2. In terms of colorectal cancer incidence in younger population, you have mentioned and cited only a study on Taiwanese population; please insert and cite data from the worldwide population, comparing briefly the differences .
  3. Please mention in text that Asian population has a different tumor behavior in colorectal cancer, therefore the findings cannot be generally applied. Please insert a paragraph as a topic of discussion.
  4. In the results you have mention a difference in the mutation findings between the right and the left colon. Please insert a paragraph that could justify these findings.
  5. Please consider to mention if the young population with the presence of mutational status has no other malignancies associated none or if not, comment that they were not screened for any other possible multiple malignancies.

The paper is valuable and I suggest its publication.

Author Response

  1. Please specify what is the international screening protocol and mention why in Taiwan the protocol is different, cite and mention comparative studies in terms of impact on the incidence.

Response from authors:

The international guideline from World Gastroenterology Organization and American Cancer Society all recommended that screening for colorectal cancer by fecal occult blood test (FOBT) and colonoscopy can start at age 50 years and continue until age 75 years. Most countries in Europe, Americas and East Asia implemented the screening program with annual or biennial FOBTs, followed by colonoscopy, when FOBT results were positive. However, due to the variation in financial resources and participation rate in different countries, the cancer detection rate varied greatly from 0.29 (Thailand) to 5.9 (Netherland) per 1000 participants (World J Gastroenterol. 2017 May 28; 23(20): 3632–3642.). In 2004, Taiwanese government launched a nationwide screening program and fecal immunochemical test (FIT) is offered biennially to individuals aged 50 to 75. This nationwide screening program has been successful, reportedly reducing CRC mortality in 5 million Taiwanese by 62% when compared screened and unscreened population. The description has been added in the first paragraph of the Introduction section in line 75-84.   

  1. In terms of colorectal cancer incidence in younger population, you have mentioned and cited only a study on Taiwanese population; please insert and cite data from the worldwide population, comparing briefly the differences.

Response from authors:

Rebecca L. Siegel had investigated the CRC incidence patterns in the United States from 1974 to 2013. They found the CRC incidence was increasing among young adults with a net increase of 4% annually compared to a net decrease of 2% annually for those age 75 years and older. (Siegel RL, Fedewa SA, Anderson WF, et al: Colorectal cancer incidence patterns in the United States, 1974-2013. J Natl Cancer Inst 109:djw322, 2017). We had inserted the information in the Introduction section in line 87-90.

  1. Please mention in text that Asian population has a different tumor behavior in colorectal cancer, therefore the findings cannot be generally applied. Please insert a paragraph as a topic of discussion.

Response from authors:

Risk factors and tumor behavior of colorectal cancer in Asia population may have some differences with Western countries (Intest Res. 2019 Jul; 17(3): 317–329), therefore the findings in this study should be carefully applied. We have added this notion in the Conclusions section in line 316-318.

  1. In the results you have mention a difference in the mutation findings between the right and the left colon. Please insert a paragraph that could justify these findings.

Response from authors:

Right side colon predominance was one of the clinical characteristics of lynch syndrome with mismatch repair gene mutations while familial colorectal cancer type X (FCCTX) with unknown mutations have left side or distal colon predominance. (British Journal of Cancer volume 111, pages598–602(2014)). This feature is concordant with our findings in which 4 of 6 sporadic early-onset CRC patients carrying MLH1 germline mutations. The description has been added to discussion section in line 223-227.

  1. Please consider to mention if the young population with the presence of mutational status has no other malignancies associated none or if not, comment that they were not screened for any other possible multiple malignancies.

Response from authors:

In theory, defects in mismatch repair genes will result in the slow and progressive somatic accumulation of pathogenic variants (two-hit theory). This will increase the risk of cancer in multiple tissues that are vulnerable to this type of damage, such as those with a rapid turnover rate (e.g., colonic and extra-enteric cells). In this study, we did not observe other malignancies among our patients instead of colorectal cancers. However, if the follow-up period is long enough, extracolonic malignancies should be monitored carefully. This description has ben added to discussion section in line 216-218.

Reviewer 2 Report

This paper evaluating the pathogenic germline mutations of DNA repair pathway components in early- onset sporadic colorectal polyp and cancer patients is well-designed work focusing genetic predisposition in sporadic CCR. The manuscript is well supported by an accurate statistical analysis along with a careful reading of the literature in the field.The authors have an original idea identifying cancer susceptible individuals by cancer multigene panel testing. Their aim was to detect cancer mutation carriers first of the screening at age 50 years with immuno-fecal blood tests (iFOBT). 142 patients enrolled were < 50 years old: 49 CRC without family history, 50 polyp patients and 49 normal coloscopy. Moreover, they found 12.2%, 4.0% and 2.3% germline cancer mutations and was according to literature. The cancer panel was well-selected with 30 CRC-associated susceptibility genes, all gatekeepers and implicated in maintenance of genome integrity as related to non-polyposis syndrome , non-adenomatous polyposis and polyposis syndrome. Interestingly, seven different mutations were detected in nine patients and all genes encode factors involved in the DNA repair pathway. This is an important instrument for precision medicine. One young sporadic CRC patients carried heterozygous mutations in both MLH1 and BRCA1. Three mutations were found each in two independent patients and were designed as “hot-spot” mutations. This study is very original because it is an opportunity to prevent mutation carriers without family history by a cost-effective genetic tool for large scale screening of the young people.

Suggestions for authors:

  • I strongly advise a careful reading of the text for the presence of many errors
  • improve english grammar, styles and punctuation
  • Three mutations in two independent patients aren’t “hot-spot” mutations, but suspect “founder” mutation of Taiwanese population. So, the authors could change the use of this term
  • It is necessary the use of HGVS nomenclature for mutation in table 2 too
  • If it is possible, the authors should indicate the dimension of these nine families with probands mutation carriers. Indeed, small families can  explain a reduced penetrance of mutation, rather than “de novo mutation”

Author Response

  1. I strongly advise a careful reading of the text for the presence of many errors improve english grammar, styles and punctuation.

Response from authors:

Thank you for your kind suggestion. The authors are not English native users. Before submission, we have corrected the possible typing or grammar error by professional English editor in Australian (www. ScienceManager.com).

  1. Three mutations in two independent patients aren’t “hot-spot” mutations, but suspect “founder” mutation of Taiwanese population. So, the authors could change the use of this term

Response from authors:

Thank you for the suggestion. We had changed the term from “ hot-spot” to suspect “founder” mutation in corresponded paragraph.

  1. It is necessary the use of HGVS nomenclature for mutation in table 2 too

Response from authors:

Yes, we have corrected the mutation in table 2 by HGVS nomenclature.

  1. If it is possible, the authors should indicate the dimension of these nine families with probands mutation carriers. Indeed, small families can explain a reduced penetrance of mutation, rather than “de novo mutation”

Response from authors:

We add a supplementary figure 1 to show the respective pedigree trees including three generation of six sporadic proband’s families. The clear no cancer history was observed in proband’s parents or even grandparent’s generation. It demonstrates the possibility of de novo mutation. The description was added in discussion section in line 182-186.